# Application of DIY Electrodermal Activity Wristband in Detecting Stress and Affective Responses of Students

**DOI:** 10.3390/bioengineering11030291

**Published:** 2024-03-20

**Authors:** Kenneth Y. T. Lim, Minh Tuan Nguyen Thien, Minh Anh Nguyen Duc, Hugo F. Posada-Quintero

**Affiliations:** 1National Institute of Education, Nanyang Technological University, Singapore 637616, Singapore; 2Independent Researcher, Singapore 357689, Singapore; nguyenthienminhtuan526705@gmail.com (M.T.N.T.); anhnguyenducminh@gmail.com (M.A.N.D.); 3Biomedical Engineering Department, University of Connecticut, Storrs, CT 06269, USA; hugo.posada-quintero@uconn.edu

**Keywords:** electrodermal activity, machine learning, maker culture, microclimate, neurological stress, physiology, thermal comfort, wearables

## Abstract

This paper describes the analysis of electrodermal activity (EDA) in the context of students’ scholastic activity. Taking a multidisciplinary, citizen science and maker-centric approach, low-cost, bespoken wearables, such as a mini weather station and biometric wristband, were built. To investigate both physical health as well as stress, the instruments were first validated against research grade devices. Following this, a research experiment was created and conducted in the context of students’ scholastic activity. Data from this experiment were used to train machine learning models, which were then applied to interpret the relationships between the environment, health, and stress. It is hoped that analyses of EDA data will further strengthen the emerging model describing the intersections between local microclimate and physiological and neurological stress. The results suggest that temperature and air quality play an important role in students’ physiological well-being, thus demonstrating the feasibility of understanding the extent of the effects of various microclimatic factors. This highlights the importance of thermal comfort and air ventilation in real-life applications to improve students’ well-being. We envision our work making a significant impact by showcasing the effectiveness and feasibility of inexpensive, self-designed wearable devices for tracking microclimate and electrodermal activity (EDA). The affordability of these wearables holds promising implications for scalability and encourages crowd-sourced citizen science in the relatively unexplored domain of microclimate’s influence on well-being. Embracing citizen science can then democratize learning and expedite rapid research advancements.

## 1. Introduction

Climate change stands as a paramount challenge of the 21st century. As affirmed by the fifth report of the Intergovernmental Panel on Climate Change (IPCC), the observable and escalating human impact on the climate system is evident, with repercussions spanning across continents and oceans [1]. Its consequences are widespread, degrading the quality of life for every creature on Earth: glaciers are receding, river and lake ice is melting prematurely, species habitats are shifting, and vegetation is blooming earlier than before [2].

A microclimate denotes a localized area within a larger surrounding area with distinct climatic conditions [3]. Consequently, any given climatic zone encompasses numerous microclimates, each varying in characteristics from the overall region. Given the general habitability of our planet, humans have populated its landmasses. However, considering the discrepancy between the human scale and the diverse habitats we inhabit, alterations in the climates of these habitats could disproportionately impact our daily activities.

As of now, the detrimental effects of anthropogenic environmental pollution have worsened the physical environment for education, teaching, and learning. For instance, heat waves have been shown to seriously impair students’ health and productivity [4]. Palme and Salvati (2021) highlighted that there is relatively inadequate research focusing on the connections between microclimates and human health and emotions [5]. Therefore, it is crucial to delve into how microclimates influence health and productivity to tackle this overlooked issue.

In June 2022, the IPCC suggested that rapidly increasing climate change presents a growing risk to mental health and psychosocial well-being, from emotional distress to anxiety, depression, grief, and suicidal behavior. Thus, the investigation of electrodermal activity (EDA)—referring to the continuous variation in the electrical characteristics of the skin, which varies with the moisture level—as a noninvasive method to detect stress and emotional arousal is an area of interest. This is especially so because productivity and stress management is an area of interest. EDA is linked to the sympathetic nervous system and consists of two components: tonic and phasic, which are represented by skin conductance level (SCL) and skin conductance response (SCR) [6]. For the frequency domain, features relative to EDASymp, TVSymp (spectral powers in specific frequency bands according to Posada-Quintero et al. (2016a; 2016b) [7,8] and their normalized versions were focused on as they were found to be highly sensitive to orthostatic, cognitive, and physical stress (Posada-Quintero et al., 2020) [9].

It is self-evident that climate change has various effects on individual well-being. As human beings, we are attuned to our immediate environment, and our reactions to shifts in microclimates can influence both emotions and health. Specifically, climate change could potentially alter microclimates to a degree where such changes could adversely affect the physical and mental well-being of those residing in these environments. For instance, a study by Liu et al. in 2019 concluded that “the increasing research interest in thermal comfort and health has heightened the need to figure out how the human body responds, both psychologically and physiologically, to different microclimates” [10]. Hence, delving into EDA data may reveal previously implicit connections regarding how microclimate correlates with our perception of well-being on a detailed scale.

Given the context of heat stress, we apply the use of EDA in the context of students’ scholastic activity. Taking on a multidisciplinary, citizen science and maker-centric approach, low-cost, bespoke wearables such as a mini weather station and biometric wristband were built. To investigate both physical health as well as stress, the instruments were first validated against research-grade devices. Following this, a research experiment was created and conducted in the context of students’ scholastic activity. Data from this experiment were used to train machine learning models, which can then interpret the relationships between the environment, health, and stress. It is anticipated that analyses of EDA data will enhance the developing model that describes the intersections between local microclimate and physiological and neurological stress.

One of the potential contributions of our research is to showcase the effectiveness and feasibility of inexpensive, self-designed wearable devices for monitoring microclimate and EDA. The affordability of these wearables holds promising implications for scaling up and, consequently, for promoting crowd-sourced citizen science in the relatively underexplored domain of microclimate and well-being relationships. To reinforce the benefits of this study, the practice of citizen science can democratize the process of learning and accelerate the progress of research [11].

## 2. Literature Review

### 2.1. Understanding Microclimate and Its Effects

The concept of microclimate, as delineated in the literature [12], pertains to the array of climatic conditions observed in localized areas close to the earth’s surface. This encompasses environmental factors such as temperature, light, wind speed, and moisture. Microclimate has played a pivotal role throughout human history, offering crucial indicators for habitat selection and various activities [13]. Regardless of the global biomes we inhabit, our bodies specifically react to microclimate conditions rather than the broader descriptors of the overall climatic region. For instance, farmers have traditionally relied on localized fluctuations in temperature and precipitation to plan their agricultural activities. Microclimate exerts a direct influence on ecological processes and manifests subtle shifts in ecosystem functioning and landscape configuration across different geographical scales [14].

Microclimate exerts an influence on electrodermal activity, which in turn correlates with our physiological and mental well-being. An illustration of this association concerning human health is evident in the impact of urban microclimate on our thermal comfort [5]. However, the connections between microclimate and biological processes are intricate and frequently nonlinear. For instance, plant distribution can be understood as a result of various factors, including light, temperature, moisture, and vapor deficit [13]. Hence, even a slight alteration in microclimate could lead to adverse effects on human emotions and health beyond just thermal comfort.

### 2.2. Urgency of Studying the Effects of Microclimates in the Context of Rapid Climate Change

The phenomenon of rapid urbanization, particularly notable in developing nations, has spurred significant migration flows toward urban centers [15]. As per Statista, global urbanization reached approximately 56% in 2020 [16]. With the pace of urbanization accelerating, alterations to urban environments and climates are inevitable [17]. This widespread urbanization complicates predictions regarding anthropogenic impacts on Earth’s climate.

At local levels, activities linked to changes in land use, land cover, and urban expansion result in various impacts, including alterations in atmospheric composition, water and energy balances, and ecosystem dynamics [18]. Given the interconnected nature of ecosystems, even minor changes in one component can trigger nonlinear effects elsewhere. For instance, a study by Xiong et al. in 2015 examined the effects of different air temperature shifts on human health and thermal comfort, revealing sensitivities such as perspiration, eye strain, dizziness, accelerated respiration, and increased heart rate as reported symptoms [19].

Amidst global climate change and the exacerbation of urban heat island effects, urban living conditions have deteriorated, significantly affecting human thermal comfort and health [10]. These effects extend beyond psychological impacts, influencing thermal sensation, mood, and concentration, to physiological repercussions such as sunburn, heat stroke, and heat cramps. Liu et al. (ibid.) have also cautioned that “global climate change and intensifying heat islands have reduced human thermal comfort and health in urban outdoor environments”.

The mentioned effects extend to the context of teaching/learning and classroom environment. For example, heat waves have been shown to seriously impair students’ health and productivity [4]. One way to explain this is that the external environmental setting can be an ambient stressor on students’ health [20]. Since climate change results in the continuous change in the external environment, e.g., the microclimate inside a classroom, a different degree of stress might be experienced by students, even teachers. With a deeper understanding of the extent to which microclimatic change can affect health, stress, and performance, measures can be taken to maximize productivity.

### 2.3. Utility of Electrodermal Activity (EDA) Data in Research

As defined by Critchley and Nagai [21], electrodermal activity (EDA) denotes a measurement of neurally mediated effects on sweat gland permeability, manifested as alterations in skin resistance to a minor electrical current or variations in electrical potential across different skin regions. EDA comprises both tonic and phasic components, represented, respectively, by skin conductance level (SCL) and skin conductance response (SCR). It is closely linked to human stress and emotional responses and can be measured noninvasively [6]. Hence, it is a viable option for reliable and accurate assessment of human stress and emotions in a more comfortable setting outside laboratories.

Ward et al. in 2004 [22] have demonstrated that electrodermal activity and heart rate variability are under the influence of the responses of the autonomic nervous system to psychological and emotional activity. They have suggested that any change in EDA during so-called sustained attention to response tests (SARTs) would also be reflected in a change in heart rate variation. As such, their work suggests the possibility of underlying correlations between electrodermal activity and other biometric factors.

EDA data have proven effective in many studies that involve emotional and stress assessment. What is more noteworthy is that there have been attempts to study EDA with a multidisciplinary approach. For example, the effects of thermal variance on a person’s electrodermal activity following the circadian rhythm were investigated by Kobas et al. [23]. Another exemplification would be the work of Fernandes et al. in linking physical and social environments with mental health using EDA data to assess emotions [24]. As such, the wide adoption of EDA data in investigating various relationships between humans and the environment means that it is a popular and suitable data type to collect.

As argued in the preceding paragraphs, changes in climate can affect individuals both physiologically and psychologically. Our review of the literature also suggests correlations between the changing conditions of the climate and changes in recorded EDA. As such, we posit that an understanding of electrodermal activity contributes to a better understanding of mental well-being with respect to changes in microclimate.

## 3. Methodology

### 3.1. Collecting Electrodermal Activity (EDA) Data

For the do-it-yourself (DIY) physiological wristband used in this study, EDA sensors are built based on the design described in Zangróniz et al. (2017) [25] with an input voltage of 3.3 V and a 10 Hz sampling rate. LM324 operational amplifier with low noise of 35 nV/rtHz was used alongside Dry Ag/AgCl Finger Electrodes. Figure 1 shows the schematic for the wristband. TP4056 battery charger circuit and 1200 mAh 3.7 V lithium battery provided approximately 10 h battery life for the unit.

Vout was measured with an external analog-to-digital converter ADS1115 with a 16-bit resolution with an internal programmable gain amplifier set at ±2.048 V maximum voltage and stored on computers or mobile devices through the HC-05 Bluetooth module. These components are secured inside a plastic container of size 6.5 cm by 5 cm by 2.5 cm. Figure 2 shows an assembled prototype.

Following Zangróniz et al. (2017) [25], skin resistance can be calculated from output voltage Vout by R_skin_ (in Ω) = (1 − 2 × V_out_/V_dd_) × R_ref_, with V_dd_ = 3.3 V, R_ref_ = 826,000 Ω, and V_out_ calculated from serial output of Arduino.

Then, skin conductance G_skin_ can be calculated by G_skin_ (in μS) = (1/R_skin_) × 10^6^.

The following are some relevant details of the DIY EDA electrodes:-ECG electrodes: Ag/AgCl coated with KCl-gel;-Diameter: 0.80 cm;-Area: 0.50 cm^2^;-Current: 1.50 μA;-Current to skin: 2.99 μA/cm^2^, below the 10 μA/cm^2^ recommendation;-Voltage to skin: around 0.30 V for well-hydrated skin, below the 0.5 V recommendation.

### 3.2. Collecting Environmental Data

To gather microclimate data, a compact portable device was constructed to assess the ambient environmental conditions, including noise level, light intensity via infrared radiation, dust concentration, carbon dioxide concentration, temperature, relative humidity, air pressure, and wind speed. The sampling rate of the unit is roughly 1 Hz. The schematics for the device are depicted in Figure 3a, and the assembled device is shown in Figure 3b.

### 3.3. Data Collection—Self-Reported Measures

Data collection through self-reported measures was carried out through an assessment of general stress and an assessment of mental stress. For the former, participants were invited to attempt a series of mathematical tasks of varying difficulty under conditions of varying stress (through manipulation of the microclimate as tasks were attempted). A baseline score was also determined before any tasks were assigned and under prevailing ambient microclimatic conditions. Participants were requested to evaluate the task difficulty on a scale ranging from 1 to 10. Similarly, they were prompted to rate their stress level using a scale from 1 to 7. The range of the scale was designed to sync with subsequent types of questionnaires for mental and workload assessment. For the purposes of the study reported in this paper, a self-reported score of task difficulty greater than 6 and a level of stress greater than 4 was interpreted as ‘high stress’; conversely, a score of task difficulty and a level of stress less than 3 was interpreted as ‘low stress’.

As for the assessment of mental stress, we chose to focus on what is generally termed acute stress, that is, short-term, event-triggered exposures to threatening or challenging stimuli that elicit a psychological and/or physiological stress response, such as delivering a public speech. To assess this, we administered the instrument known as the Self-Assessment Manikin (SAM) to participants. SAM is a “non-verbal pictorial assessment technique that directly measures the pleasure, arousal, and dominance associated with a person’s affective reaction to a wide variety of stimuli” [26]. SAM was chosen for its effectiveness in directly assessing the pleasure, arousal, and dominance associated with response to the event, which reduces the resources needed to measure other types of variables for higher resolution. Furthermore, SAM can be used universally as language barriers can be overcome when emotions are represented by pictures.

#### 3.3.1. General State Assessment

Participants rated the difficulty of the task they had just carried out on a scale of 1 to 10 and the stress level on a scale of 1 to 7.

For this experiment, high-stress windows are chosen as windows with subjective task difficulty rated ≥ 7 (1 to 10) and subjective stress level rated ≥ 5 (1 to 7); low-stress windows are windows with subject task difficulty and subjective stress level rated ≤ 2 and baseline windows. These ranges were determined so that stress levels are significantly different between high-stress and low-stress windows.

#### 3.3.2. Mental State Assessment

For stress in this study, we focused on acute stress—short-term, event-based exposures to threatening or challenging stimuli that evoke a psychological and/or physiological stress response, such as giving a public speech. Before the participants started to perform a mathematical task, they were asked to fill out a pre-task survey. In the same manner, a post-task survey would be filled in after the duration of the task was allocated. For the purposes of this study, there are 4 types of questionnaires combined to assess the mental state of participants.

Firstly, Self-Assessment-Manikin (SAM) was used. As mentioned before, SAM is a non-verbal pictorial assessment technique that directly measures the pleasure, arousal, and dominance associated with a person’s affective reaction to a wide variety of stimuli. SAM was included in both the pre-task survey and the post-task survey.

Secondly, a short version of the State-Trait Anxiety Inventory (STAI) was used. STAI is a self-report scale that assesses separate dimensions of “state” and “trait” anxiety. The essential qualities evaluated by the STAI are feelings of apprehension, tension, nervousness, and worry. Scores on the STAI S-Anxiety scale increase in response to physical danger and psychological stress and decrease as a result of relaxation training [27]. Table A1 in Appendix A shows STAI questions used in the post-task survey.

Thirdly, the acute stress appraisal questionnaire was used. The acute stress appraisal emphasizes the multifaceted nature of demand and resource appraisals to be used in laboratory stress paradigms. Demands were defined to be made up of perceived uncertainty, required effort, and how demanding the task seems, among other factors, whereas resources comprise perceived knowledge and abilities, controllability, social support, and expectations [28]. There are two parts to this questionnaire: a pre-task appraisal (Table A2a) and a post-task appraisal (Table A2b). Refer to Appendix A for these tables of questionnaires.

Finally, to assess mental workload, a NASA Task Load Index (TLX) questionnaire was also included in the post-task survey. NASA-TLX is a multidimensional scale designed to obtain workload estimates from one or more operators while they are performing a task or immediately afterward. The use of NASA-TLX has gone beyond the aviation field, showing how popular and reliable it is as a means to measure workload [29].

### 3.4. Validation of EDA Data of DIY EDA Sensors against Empatica E4 EDA Sensors

In order to validate the legitimacy of the DIY EDA signals, an experiment was modeled after Zangróniz et al. (2017) [25] and Atkins et al. (2019) [30]. It was conducted within a supervised environment in which the fan, the light, and the door could all be independently controlled.

Empatica E4 (4 Hz sampling rate with silver (Ag) plated with metallic core electrodes), a research-grade equipment for measuring EDA data, was used to compare against the performance of DIY devices. Various stimuli were investigated during a session wherein the DIY device was worn on the left wrist while the Empatica E4 was worn on the right wrist. First, a sudden visual stimulus with high arousal and low valence in the OASIS database was introduced at a random time to startle the human subject. This was repeated four times consecutively. For the experiment’s second part, ten pictures that were labeled with high arousal and low valence in the OASIS database were shown consecutively for six seconds each. Blank images with a fixed duration of one second were inserted before each picture. Afterward, the subject was given a geographical task to neutralize his emotional state. Subsequently, ten pictures that were labeled with low arousal and high valence in the OASIS database were shown. Blank images were also inserted. Then, a different geographical task was assigned to return the emotional state to neutral.

The EDA data underwent initial resampling to a 4 Hz sampling rate. Subsequently, each set of resampled data was filtered using a 32nd-order Butterworth low-pass filter with a cutoff frequency of 1.5 Hz to eliminate artifacts. Following this, the data were normalized and smoothed using a moving average.

Figure 4a,b show two sample plots of EDA data traces from the wristband (worn on left wrist) compared to Empatica E4 (worn on right wrist). Visually, the phasic and tonic components extracted from both DIY EDA and E4 EDA signals using convex optimization methods for EDA (cvxEDA) demonstrate strong congruence, as indicated by high Spearman coefficients (ρ) in Figure 5a,b.

### 3.5. Investigating How the Environment Affects Physiological, Mental Health, and Productivity

The two-hour period was divided into eight 15 min intervals/windows. Each interval featured varied, randomized combinations of microclimatic factors, as illustrated in Figure 6. Participants wore a DIY wristband on the wrist of their non-dominant arm throughout the experiment while engaging in challenging mathematical tasks aimed at sustaining elevated stress levels (refer to Figure 6 for a photo of a participant during the experiment). Few windows had easier tasks to serve as baseline and low-stress periods. The participants should be junior college students with similar levels of mathematical competency. There are also baseline periods/breaks before, during, and after the experiment.

The collected data underwent processing in Python, with outlier detection conducted using the z-score method to eliminate outlier data. Initially, the collected EDA data were normalized and filtered using a low-pass filter (1.5 Hz, Butterworth, 32nd order) to eliminate unwanted artifacts [31]. Subsequently, the EDA data were decomposed into tonic and phasic components utilizing the convex optimization (cvxEDA) method developed by Greco et al. in 2016 [32]. The skin conductance level (SCL) index was derived as the mean of 2 min windows of the tonic component, while the number of skin conductance responses per minute (NSSCR) represented the quantity of rapid transient events within the phasic component [33].

According to the literature, NSSCR refers to skin conductance responses occurring without a specific eliciting stimulus [34], and NSSCR levels increase alongside ratings of emotional arousal [35]. As per Wichary et al. (2016), heightened arousal and negative valence characterize emotional stress [36]. Thus, this metric serves as a potentially reliable indicator of stress.

To conduct frequency-domain analysis, the EDA data were downsampled to 2 Hz. Subsequently, the signals underwent high-pass filtering (0.01 Hz, Butterworth, 8th order) to eliminate any underlying trend. Using variable frequency complex demodulation, TVSymp is calculated as the mean of time-varying spectral amplitudes in the 0.08–0.24 Hz band, and modified TVSymp (MTVSymp) is obtained as the difference between current TVSymp and the mean value of TVSymp of the previous 5-s window [8]. Negative values of MTVSymp were set to 0.

In the frequency domain, a Blackman window (length of 128 points) was applied to each segment (0.5-s overlap with each other), and the fast Fourier transform was calculated for each windowed segment. EDASymp(n), as a tool for sympathetic tone assessment, was computed as the normalized power within the frequency band of interest (0.045 to 0.25 Hz) [33].

The differential feature, dphEDA, is computed as the derivative of the phasic component of EDA.

The features, such as 2 Hz signals, are then synced with environmental data. Spearman correlation, appropriate for non-normally distributed data, is used to assess monotonic associations between environmental factors and EDA features.

Random forest regression (RF) models, support vector machines (SVM), and linear regression (LR) models were trained. Each type of model was trained via 2 methods each: Leave one subject out and 70% train—30% test. The mean cross-validation $R^{2}$ score was compared to see which models performed the best. Hence, as seen in Table 1, random forest regression models perform the best. Thus, it was chosen as the tool of analysis.

Random forest regression models are then trained on environmental data and EDA features, with the former as input and the latter as output, with a train–test split ratio of 7:3 to discover the nonlinear relationships between environmental factors and EDA features. The outcomes of the random forest regression models were analyzed using Shapley values and Shapley summary plots to uncover intricate connections between input and output variables.

## 4. Results

In total, more than 30,000 environmental data points and 300,000 EDA data points were collected from five participants (four males and one female).

### 4.1. Preliminary Statistical Analysis

To help with explainability and data evaluation, Shapley values were used. Shapley values were used to measure the contributions of input features to the output of a machine learning model at the instance level. Thus, Shapley values were used to interpret ML models to explain the impact of environmental factors on the output [37].

The features were then synced with environmental data. For each feature, we first performed the Kolmogorov–Smirnov normality test (we calculated the Fisher’s ratio as none of the features displayed normal distribution). Spearman correlation, appropriate for non-normally distributed data, was subsequently used to assess monotonic associations between environmental factors and EDA features.

Random forest regression models, linear regression models, and SVM on the environmental data and EDA features were trained. We compared these various models using leave-one-subject-out cross-validation. From this comparison, random forest emerged as the most accurate.

With a view to explainability—the exploration of which environmental factors significantly affect EDA features and in what way—random forest regression models were then trained on environmental data and EDA features with the former as input and the latter as output. The train–test split ratio was 7:3 to find the non-linear connections between the environmental factors and EDA features.

Figure 7 shows the box and whisker plot for different EDA feature’s statistics. Referring to the aforementioned methodology, number 1 on the *x*-axis represents high-stress events/difficult tasks, and number 2 on the *x*-axis represents low-stress events/easy tasks and baseline. Table 2 compares the EDA features during low-stress events and high-stress events.

Table 3 presents the correlations between the self-reported (subjective) data. We observed a strong positive correlation between post-task stress and post-task arousal. We also observed moderate negative correlations between post-task stress and post-task valence, as well as between post-task arousal and post-task valence. Generally, high stress correlates with low valence and high arousal. Finally, as might be reasonably expected, there is a moderate positive correlation between stress and task difficulty.

Correlations between subjective data using Spearman (non-normal distribution):-Post-task Stress and Arousal have strong positive correlation.-Post-task Stress and Valence have moderate negative correlation.-Post-task Arousal and Valence have moderate negative correlation.-High Stress correlates with low Valence and high arousal.-Post-task Stress and task difficulty have moderate positive correlation.

Table 4 presents the correlations between the environmental data and the EDA data during high-stress events.

Table 5 presents the correlations between the subjective data and the environmental data during high-stress events.

### 4.2. Predicting Subjective Data Using Random Forest Regressor with Environmental Variables as Inputs for High-Stress Windows

The results of the random forest regression models were interpreted using Shapley values and Shapley summary plots to find more complex relationships between input and output. Figure 8, Figure 9, Figure 10, Figure 11, Figure 12, Figure 13, Figure 14, Figure 15, Figure 16, Figure 17 and Figure 18 present the results of this analysis.

#### 4.2.1. Result of Random Forest Regressor on Post Arousal

From Figure 8a,b, the Shapley summary plot suggests that the post arousal is most significantly affected by the variable ‘Pressure’, followed by the variables ‘Temperature’ and ‘Humidity’. For the variable ‘Pressure’, higher pressure tends to increase emotional arousal. For the variable ‘Temperature’, higher temperature tends to decrease emotional arousal. Finally, for the variable ‘Humidity’, higher humidity tends to decrease emotional arousal.

#### 4.2.2. Result of Random Forest Regressor on Post-Stress

From Figure 9a,b, the Shapley summary plot suggests that the post-stress is most significantly affected by the variable ‘Pressure’, followed by the variables ‘Humidity’ and ‘Temperature’. For the variable ‘Pressure’, higher pressure tends to increase stress. For the variable ‘Humidity’, lower values of humidity tend to increase stress. Finally, for the variable ‘Temperature’, higher temperature tends to decrease stress.

#### 4.2.3. Result of Random Forest Regressor on Post Valence

From Figure 10a,b, the Shapley summary plot suggests that the post valence is most significantly affected by the variable ‘Pressure’, followed by the variables ‘Temperature’ and ‘CO_2_ concentration’. For the variable ‘Pressure’, higher pressure tends to decrease emotional valence. For the variable ‘Temperature’, a higher temperature can either increase or decrease emotional valence. Finally, for the variable ‘CO_2_ concentration’, higher carbon dioxide concentration tends to increase emotional valence.

### 4.3. Predicting EDA Features Using Random Forest Regressor with Environmental Variables as Inputs for High-Stress Windows

#### 4.3.1. Result of Random Forest Regressor on Tonic Component (SCL)

From Figure 11a,b, the Shapley summary plot suggests that the tonic component of EDA is most significantly affected by the variable ‘Carbon dioxide concentration’, followed by the variable ‘Temperature’. For the variable ‘Carbon dioxide concentration’, higher carbon dioxide concentration tends to decrease tonic value, which in turn correlates with the higher stress levels of participants. For the variable ‘temperature’, higher temperature also decreases tonic value, which in turn also correlates with higher stress levels.

#### 4.3.2. Result of Random Forest Regressor on Time-Varying Index of Sympathetic Activity (TVSymp)

From Figure 12a,b, the Shapley summary plot suggests that TVSymp is most significantly affected by the variable ‘CO_2_ concentration’, followed by the variables ‘Temperature’ and ‘Humidity’. For the variable ‘CO_2_ Concentration’, a higher value of CO_2_ concentration tends to decrease the TVSymp value, which results in lower stress levels and emotional valence. For the variable ‘Temperature’, a higher temperature can either increase or decrease the value of TVSymp. Therefore, it is inconclusive to which extent a change in temperature may affect stress. Finally, for the variable ‘Humidity’, lower humidity can either increase or decrease the value of TVSymp.

#### 4.3.3. Result of Random Forest Regressor on Non-Specific Skin Conductance Responses (NSSCR)

From Figure 13a,b, the Shapley summary plot suggests that the NSSCR is most significantly affected by the variable ‘Carbon dioxide concentration’, followed by the variables ‘Infrared radiation’ and ‘temperature’. For the variable ‘Carbon dioxide concentration’, higher carbon dioxide concentration tends to decrease NSSCR value, which in turn correlates with lower stress levels and emotional arousal of participants. For the variable ‘Infrared radiation’, it can be seen that a higher level of infrared radiation (intensity) leads to a higher NSSCR value and vice versa. Thus, higher infrared radiation can lead to higher levels of stress and emotional arousal.

#### 4.3.4. Result of Random Forest Regressor on Time-Invariant Spectral Index of EDA (EDASymp [μS2])

From Figure 14a,b, the Shapley summary plot suggests that the EDASymp is most significantly affected by the variable ‘Temperature’, followed by the variables ‘Carbon dioxide concentration’ and ‘Humidity’. For the variable ‘Temperature’, higher temperature/lower temperature can either increase or decrease the EDASymp value. For the variable ‘Carbon dioxide concentration’, it can be seen that higher carbon dioxide concentration tends to increase the EDASymp value.

### 4.4. Machine Learning Models Using EDA Data Features as Input and Environmental Variables as Output (Reverse Model)

#### 4.4.1. Result of Random Forest Regressor on Temperature

From Figure 15a,b, the Shapley summary plot suggests that lower temperature is related to high values of tonic_mean. For the variable ‘NSSCR’ (NSwindow), higher temperature is suggested to be related to low values of NSSCR. For the variable ‘TVSymp’, a lower temperature is suggested to be related to low values of TVSymp. For the variable ‘EDASymp’, a lower temperature is suggested to be related to high values of EDASymp.

#### 4.4.2. Result of Random Forest Regressor on Carbon Dioxide Concentration

From Figure 16a,b, the Shapley summary plot suggests that lower carbon dioxide concentration is related to high values of tonic_mean. For the variable ‘NSSCR’ (NSwindow), higher carbon dioxide concentration is suggested to be related to high values of NSSCR. For the variable ‘TVSymp’, lower carbon dioxide concentration is suggested to be related to high values of TVSymp. For the variable ‘EDASymp’, higher carbon dioxide concentration is suggested to be related to high values of EDASymp.

#### 4.4.3. Result of Random Forest Regressor on Humidity

From Figure 17a,b, the Shapley summary plot suggests that a lower value of humidity is related to high values of tonic_mean. For the variable ‘NSSCR’ (NSwindow), higher humidity is suggested to be related to low values of NSSCR. For the variable ‘TVSymp’, lower humidity is suggested to be related to low values of TVSymp. For the variable ‘EDASymp’, lower humidity is suggested to be related to high values of EDASymp.

#### 4.4.4. Result of Random Forest Regressor on Infrared Radiation (IR)

From Figure 18, the Shapley summary plot suggests that a lower value of IR is related to low values of tonic_mean. For the variable ‘NSSCR’ (NSwindow), it is inconclusive whether a higher or lower value of IR is related to high or low values of NSSCR. For the variable ‘TVSymp’, higher humidity is suggested to be related to low values of TVSymp. For the variable ‘EDASymp’, higher humidity is suggested to be related to high values of EDASymp.

Across preliminary findings by using the Spearman correlation coefficient test and findings from machine learning of random forest regression, temperature has the greatest impact on cognitive stress, for being extremely influential in drastically changing electrodermal activity, an indicator of cognitive stress. Furthermore, it was found that air quality also has a great impact on participants’ cognitive stress.

It is acknowledged that the approach in this report has some limitations. There might be a degree of uncertainty in data measurements made by self-built devices and a limited number of participants. However, it was overcome by a two-prong approach to confirm the reliability of the suggested findings. A two-way analysis was used, that is, reversing the roles of EDA feature data and environmental data for machine learning models, in combination with preliminary findings from the Spearman correlation coefficient.

## 5. Discussion

Across preliminary findings by using the Spearman correlation coefficient test and findings from machine learning of random forest regression, temperature has the greatest impact on cognitive stress, for being extremely influential in drastically changing electrodermal activity, an indicator of cognitive stress. Overall, temperature tends to increase cognitive stress but makes emotional arousal and emotional valence more negative (less positive). Furthermore, it was found that air quality also has a great impact on participants’ cognitive stress.

This has demonstrated a universal approach of using the Shapley summary plot to interpret both the direction and magnitude of the effects of microclimate on EDA features. To elaborate, the study has shown that based on the magnitude of Shapley values, the significance and extent to which microclimate affects cognitive stress and emotional states can be inferred. With a known effect of increasing or decreasing the value corresponding to the EDA feature, literature can be reviewed to understand how cognitive stress and emotional state change accordingly.

The methodology of the project was designed to the best of its ability to ensure the change in cognitive stress is due to the change in microclimate and difficult math tasks. This was due to the randomization of different climatic conditions and the alternation of easy mathematical tasks among very challenging mathematical tasks. Furthermore, this experiment was designed to be as short as possible to minimize mental fatigue, as well as to ensure EDA is not affected significantly by participants’ circadian rhythm.

It is acknowledged that the approach in this report has some limitations. There might be a degree of uncertainty in data measurements made by self-built devices and a limited number of participants. However, it was overcome by a two-prong approach to confirm the reliability of the suggested findings. A two-way analysis was used, that is, reversing the roles of EDA feature data and environmental data for machine learning models, in combination with preliminary findings from the Spearman correlation coefficient. Another key consideration for the future is to consider the significance of circadian rhythm, as it can also affect EDA data.

## 6. Concluding Remarks

This study has demonstrated the feasibility of DIY, citizen science electronic wearables in research on physiological well-being. Not only are the DIY, low-cost wearables’ data collected comparable to that of high-end lab-grade equipment, but they also performed exceptionally well in providing a high enough resolution of data. This was shown consistently through the results of analysis using machine learning models and literature review.

The nature of DIY electronic wearables also has implications for its scale of usage and application. At a significantly lower cost, this device can be produced at a larger scale, modified, and used almost instantly for various purposes of research. This can better drive the trend of citizen science, where the quality of results is not as emphasized as the process of carrying out research. With this, it is hoped that our wearable will be able to provide a means of conducting research, democratizing the equity of research to everyone and, hence, accelerating the overall progress in the field.

The research has also demonstrated the success of utilizing a multi-model and multidisciplinary approach to understanding the link between microclimate and human health, stress, and emotions. Thus, an extension of the research can be to introduce photoplethysmography (PPG) data into the multi-model approach. By nature, PPG data are nonintrusive to measure and also has been shown to effectively detect and manage stress levels. Coincidentally, PPG can also be measured most effectively at the wrist of participants, making it a consideration for future expansion of the biometric wristband.

To improve the mentioned limitations, in the future, the participants’ pool can be increased to ensure genders are equally represented. More participants also ensure more statistical power and account for a wider range of physiological variability. More trends can then be generalized. In terms of methodology, mathematical tasks can be replaced with other activities, such as learning, to investigate other factors, such as retention rate.

With further application of such electronic wearables, it is hoped that in the near future, better solutions can be developed to maximize comfort and productivity, not just in the quasi-formal academic context. This can be in the form of redesigning infrastructure and biomes. This study hopes to set an example for future research to expand and explore using a more robust, comprehensive approach (e.g., including more environmental factors and/or a multi-modal approach using PPG and EEG).

## Figures and Tables

**Figure 1 bioengineering-11-00291-f001:**
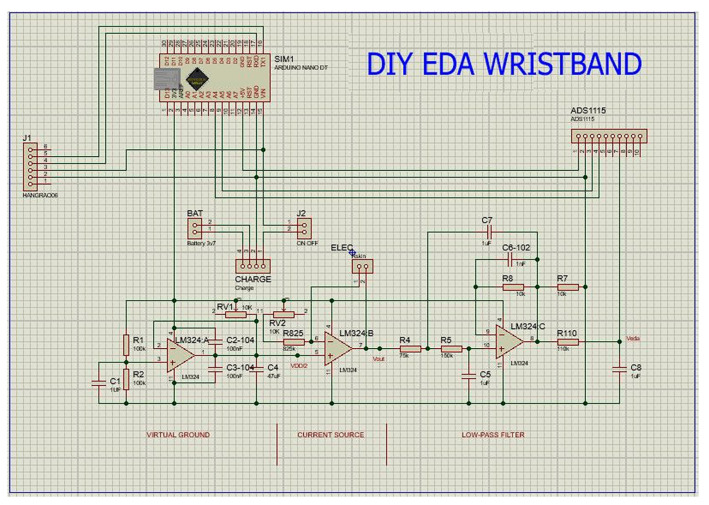
Schematic of EDA wristband.

**Figure 2 bioengineering-11-00291-f002:**
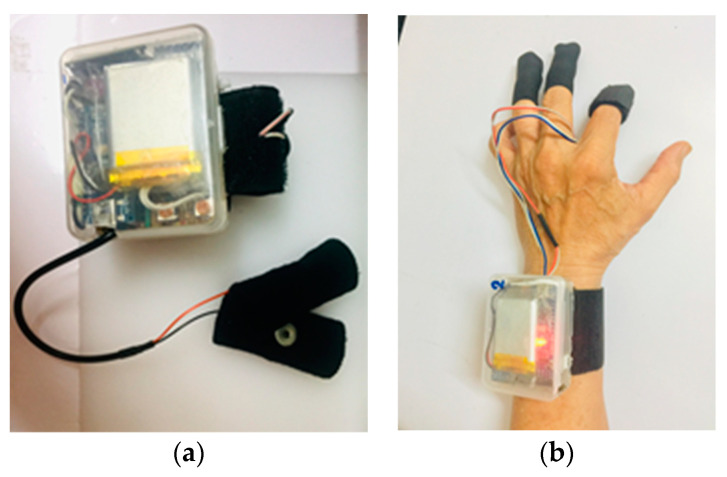
(**a**,**b**) Assembled EDA wristband.

**Figure 3 bioengineering-11-00291-f003:**
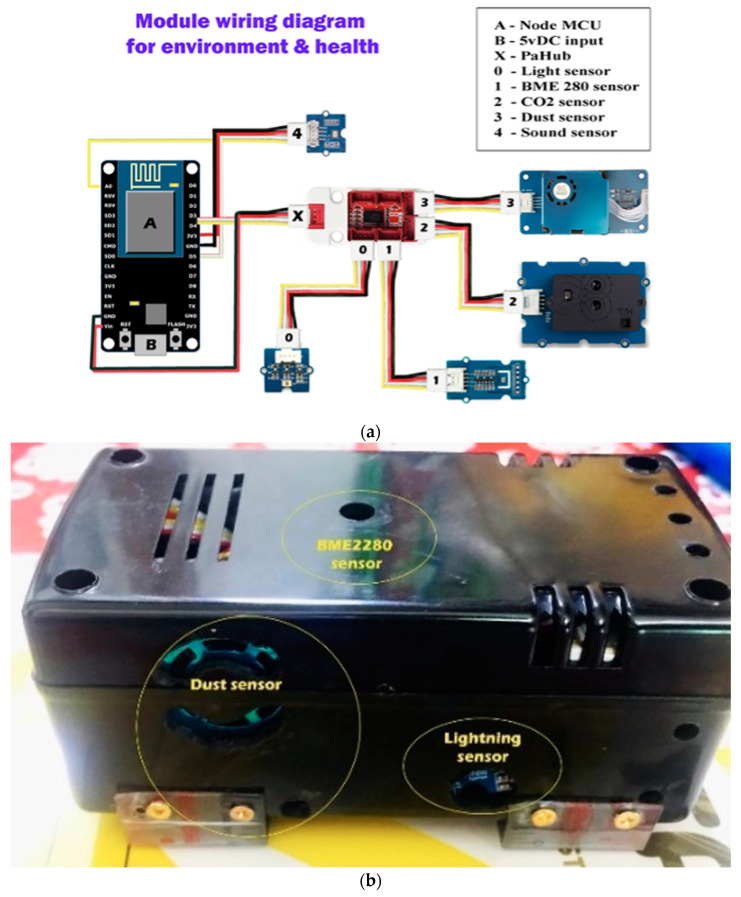
(**a**,**b**) Schematics for environmental sensor device and assembled device.

**Figure 4 bioengineering-11-00291-f004:**
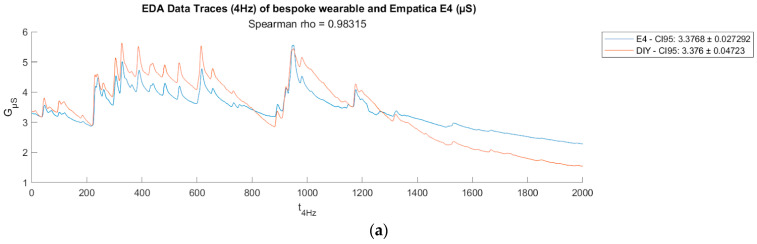
(**a**) Comparison of raw EDA data traces (ρ = 0.983),(**b**) Comparison of raw EDA data traces from another participant (ρ = 0.813).

**Figure 5 bioengineering-11-00291-f005:**
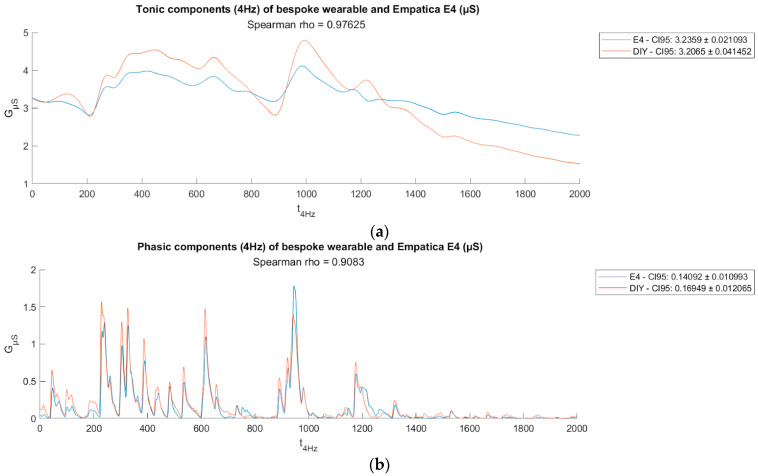
(**a**) Comparison of tonic components (n.u., ρ = 0.976); (**b**) comparison of phasic components (n.u., ρ = 0.874).

**Figure 6 bioengineering-11-00291-f006:**
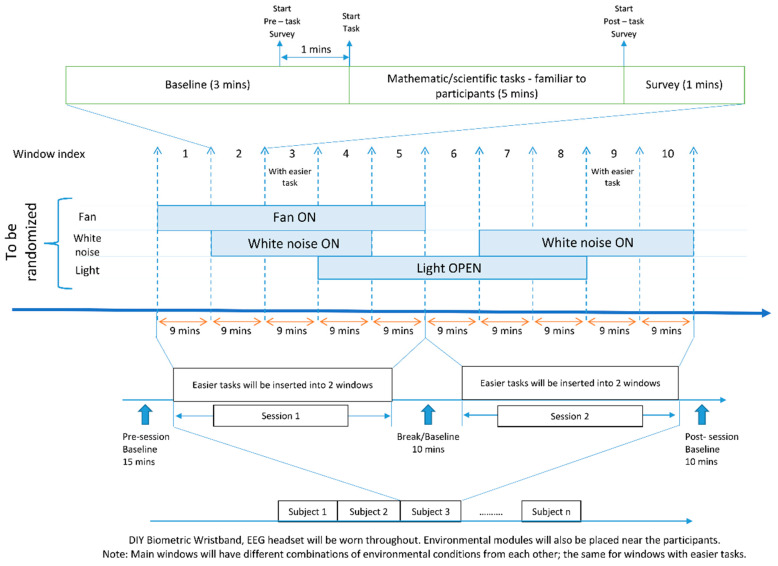
Standard of procedure of testing DIY wearable device in a controlled environment experiment.

**Figure 7 bioengineering-11-00291-f007:**
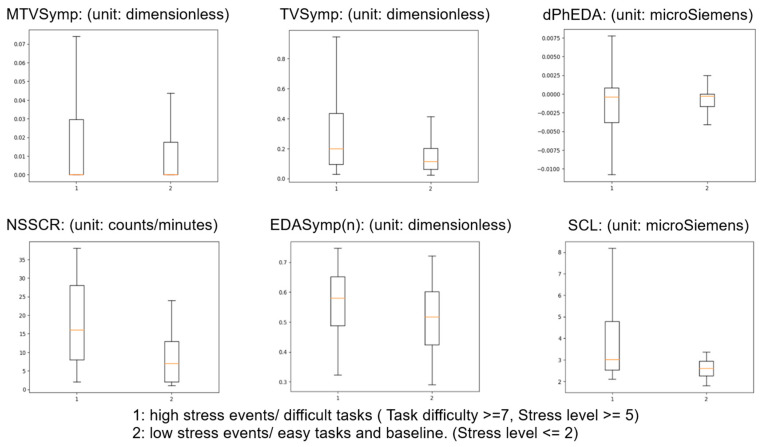
Box and whisker plot of statistics of EDA features.

**Figure 8 bioengineering-11-00291-f008:**
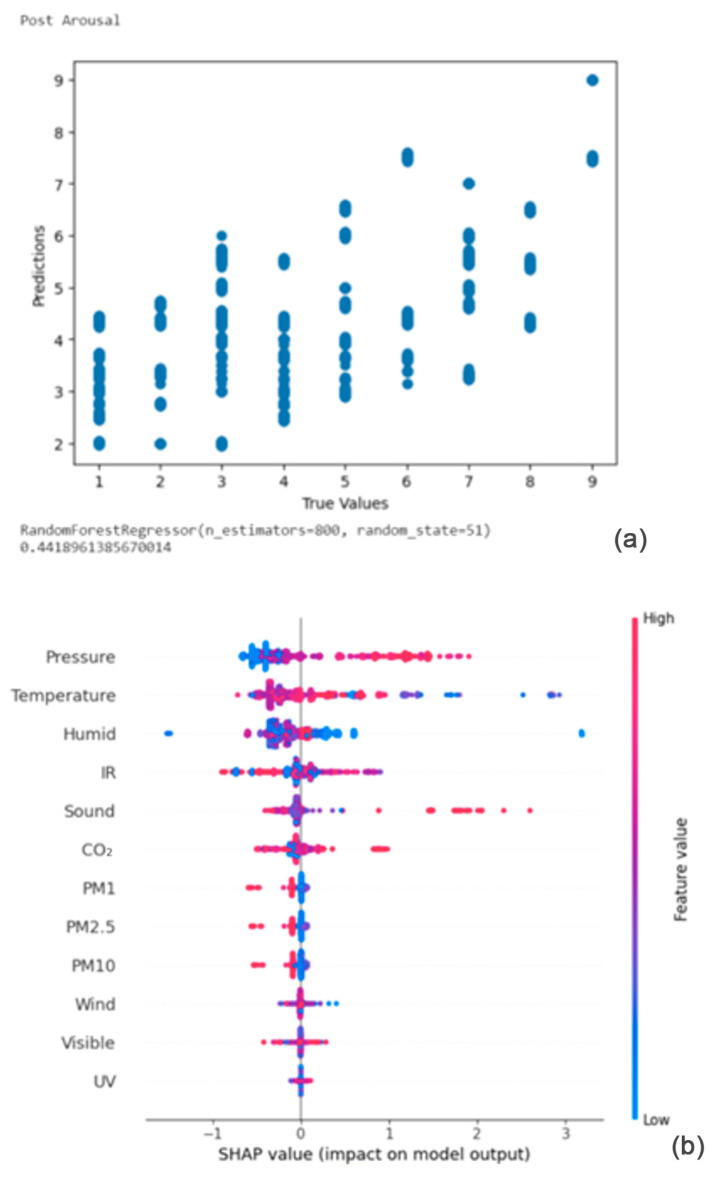
(**a**,**b**) $R^{2}$ Score of 0.442 and Shapley summary plot using environmental data as input to predict post arousal.

**Figure 9 bioengineering-11-00291-f009:**
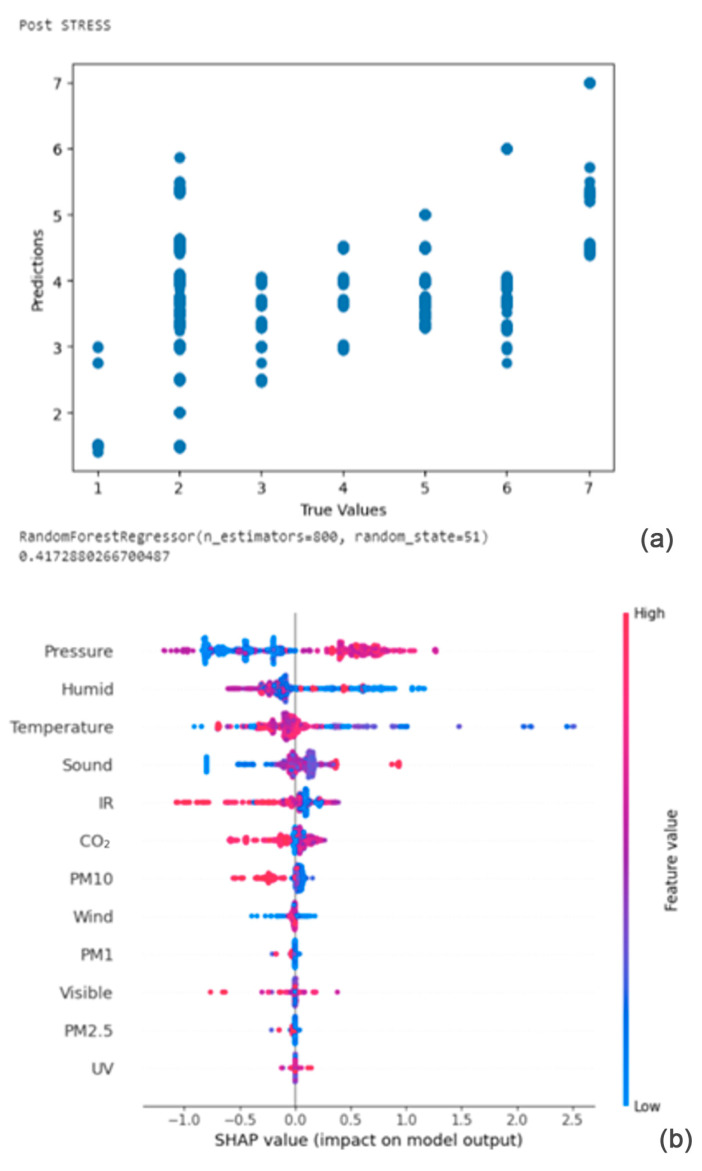
(**a**,**b**) $R^{2}$ Score of 0.417 and Shapley summary plot using environmental data as input to predict post-stress.

**Figure 10 bioengineering-11-00291-f010:**
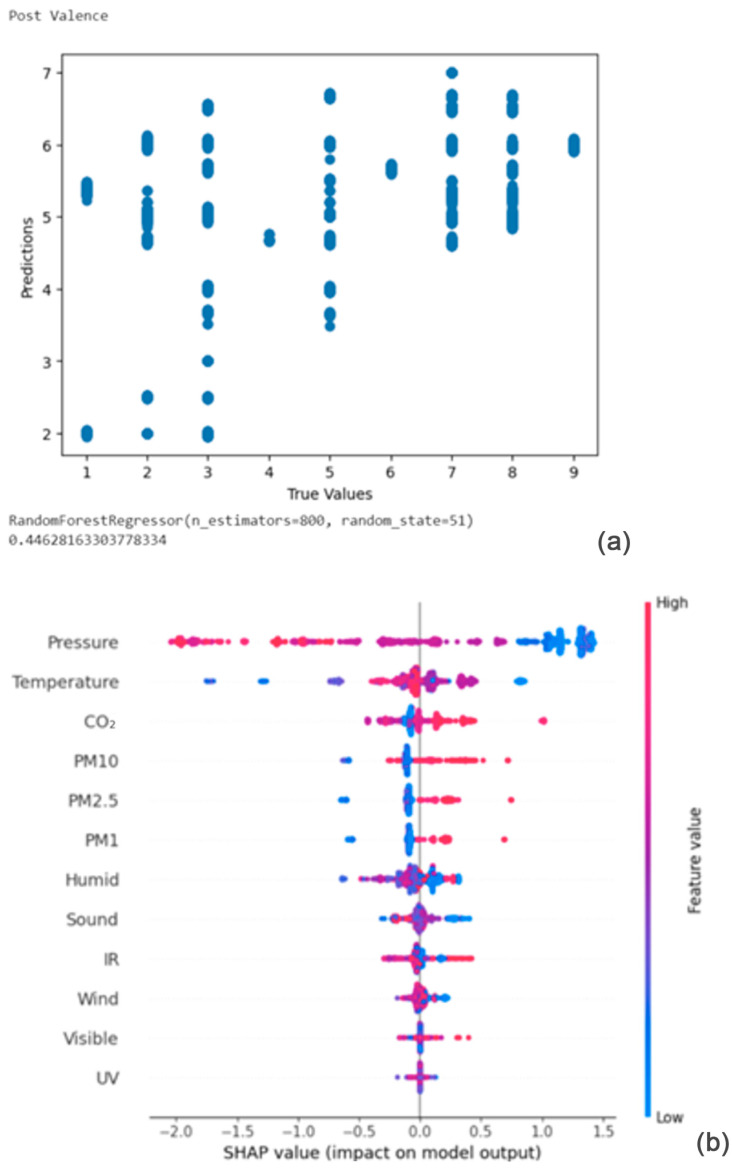
(**a**,**b**) $R^{2}$ Score of 0.446 and Shapley summary plot using environmental data as input to predict post valence.

**Figure 11 bioengineering-11-00291-f011:**
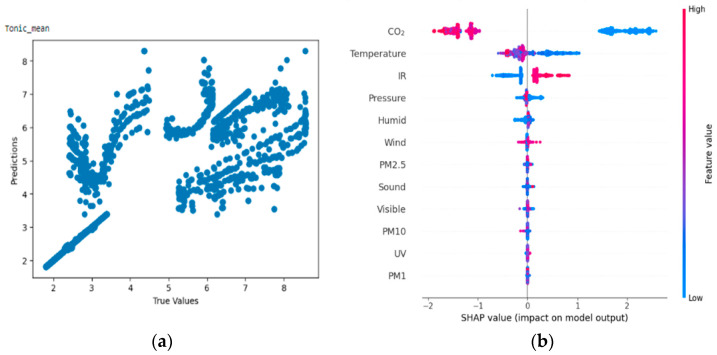
(**a**,**b**) $R^{2}$ Score of 0.673 and Shapley summary plot using environmental data as input to predict tonic_mean.

**Figure 12 bioengineering-11-00291-f012:**
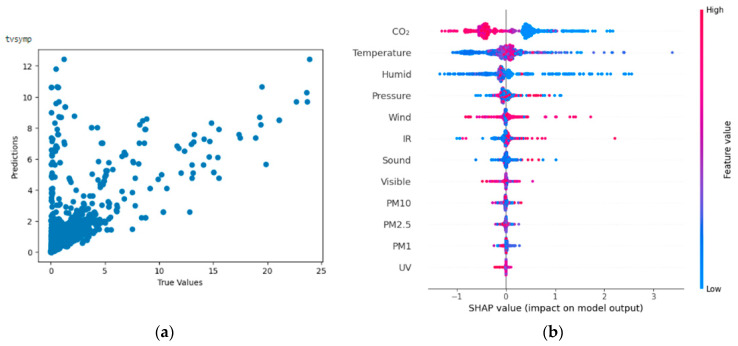
(**a**,**b**) $R^{2}$ Score of 0.425 and Shapley summary plot using environmental data as input to predict TVSymp.

**Figure 13 bioengineering-11-00291-f013:**
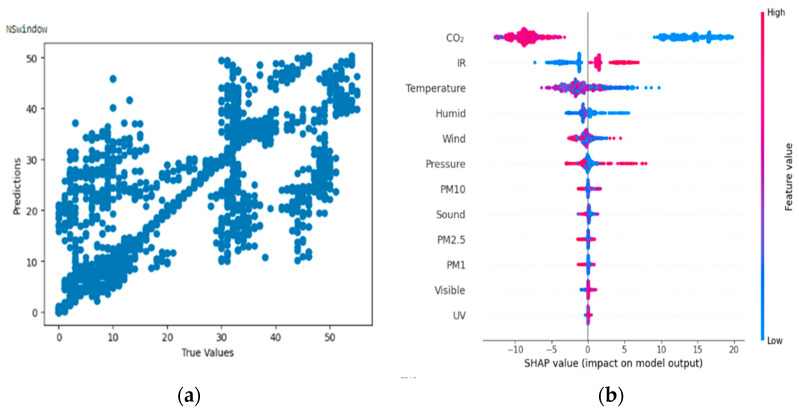
(**a**,**b**) $R^{2}$ Score of 0.622 and Shapley summary plot using environmental data as input to predict NSSCR.

**Figure 14 bioengineering-11-00291-f014:**
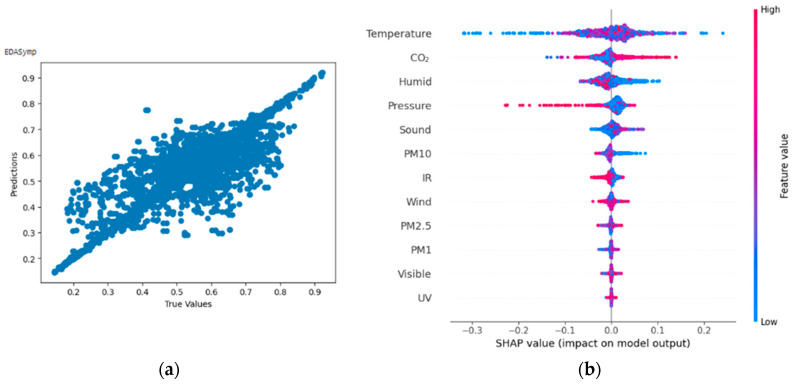
(**a**,**b**) $R^{2}$ Score of 0.740 and Shapley summary plot using environmental data as input to predict EDASymp.

**Figure 15 bioengineering-11-00291-f015:**
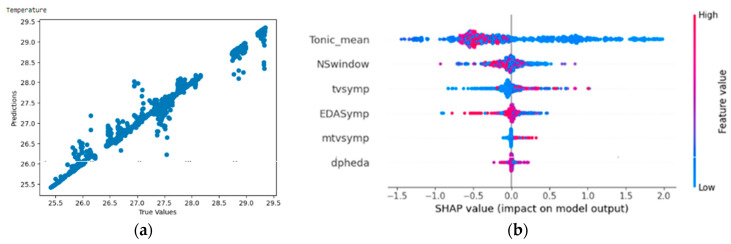
(**a**,**b**) $R^{2}$ Score of 0.982 and Shapley summary plot using EDA data features as input to predict temperature.

**Figure 16 bioengineering-11-00291-f016:**
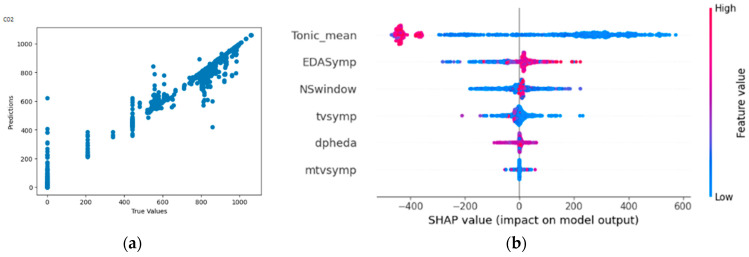
(**a**,**b**) $R^{2}$ Score of 0.987 and Shapley summary plot using EDA data features as input to predict carbon dioxide concentration.

**Figure 17 bioengineering-11-00291-f017:**
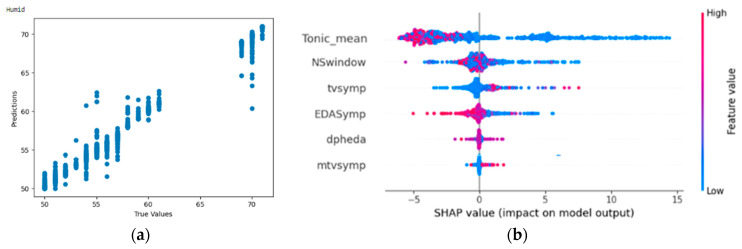
(**a**,**b**) $R^{2}$ Score of 0.986 and Shapley summary plot using EDA data features as input to predict humidity.

**Figure 18 bioengineering-11-00291-f018:**
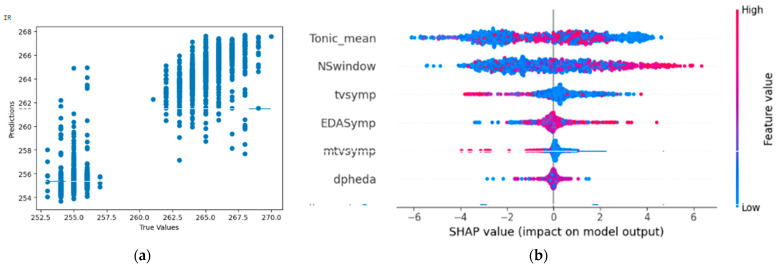
(**a**,**b**) $R^{2}$ Score of 0.919 and Shapley summary plot using EDA data features as input to predict IR.

**Table 1 bioengineering-11-00291-t001:** $R^{2}$ score for each type of machine learning model on various EDA features.

$\text{Mean Cross-Validation}\;R^{2}$	MTVSymp	TVSymp	dPhEDA	Nswindow (NSSCR)	EDASymp	Tonic_Mean (SCL)
Leave One Subject Out—RF	0.260	0.296	−0.077	0.366	0.373	0.392
70% Train—30% Test—RF	0.360	0.411	0.146	0.604	0.736	0.688
Leave One Subject Out—SVM	−0.054	−0.186	−1.039	−0.557	−0.470	−0.460
70% Train—30% Test—SVM	0.000	0.038	−0.240	0.574	0.205	0.680
Leave One Subject Out—LR	−0.142	−0.203	−0.020	−0.865	−0.280	−2.800
70% Train—30% Test—LR	0.033	0.083	0.000	0.562	0.054	0.687

**Table 2 bioengineering-11-00291-t002:** Comparing EDA features during low-stress events and high-stress events.

Feature	MTVSymp	TVSymp	dPhEDA	NSSCR	EDASymp	SCL
Mean of Fisher’s Ratio	0.2488	0.4988	0.07091	0.7122	0.3265	0.4702
Max of Fisher’s Ratio	0.4305	0.8206	0.1236	1.490	0.7166	0.5988
Mean ± STD during high stress	0.02547 ± 0.04754	0.3216 ± 0.3141	−0.001583 ± 0.01091	17.65 ± 11.15	0.5662 ± 0.1082	4.103 ± 2.637
Mean ± STD during low stress	0.01251 ± 0.02116	0.1535 ± 0.1219	−0.0007863± 0.002670	8.394 ± 6.674	0.5155 ± 0.1110	2.808 ± 0.7934

**Table 3 bioengineering-11-00291-t003:** Correlations between subjective data (non-normal distribution) using Spearman’s rank correlation.

	Post Difficulty	Post-Stress	Post Dominance	Post Arousal	Post Valence
Post Difficulty	1.000	0.498	−0.233	0.379	−0.366
Post-Stress	0.498	1.000	−0.173	0.784	−0.431
Post Dominance	−0.233	−0.173	1.000	−0.219	0.653
Post Arousal	0.379	0.787	−0.219	1.000	−0.352
Post Valence	−0.366	−0.431	0.653	−0.352	1.000

**Table 4 bioengineering-11-00291-t004:** Correlations between environmental data and EDA data during high-stress events using Spearman’s rank correlation.

	MTVSymp	TVSymp	DphEDA	NSSCR	EDASymp	SCL
Sound	−0.05934	−0.32213	0.06365	−0.40921	0.05706	−0.61090
Visible	−0.00873	0.01804	−0.01563	0.14515	0.04325	0.03541
IR	0.01359	0.04459	−0.02861	0.21509	0.05548	0.08022
UV	−0.00863	0.01278	−0.01280	0.13655	0.04347	0.02383
Temperature	−0.03886	−0.14755	0.01628	−0.26133	−0.10101	−0.50281
Humid	−0.05476	−0.23905	0.04124	−0.35894	−0.22702	−0.40194
Pressure	−0.03686	−0.23548	0.05704	−0.28958	−0.05870	−0.32578
CO_2_	−0.10350	−0.46585	0.08311	−0.54133	0.03551	−0.78979
PM_1_	−0.06542	−0.33744	0.08212	−0.47146	−0.12212	−0.60179
PM_2.5_	−0.05987	−0.33496	0.08100	−0.47356	−0.13784	−0.60737
PM_10_	−0.04861	−0.33464	0.08080	−0.45493	−0.10840	−0.66254
Wind	0.02467	0.16250	−0.03055	0.19242	0.04873	−0.05900

**Table 5 bioengineering-11-00291-t005:** Correlations between subjective data and environmental data during high-stress events using Spearman’s rank correlation.

	Post Difficulty	Post-Stress	Post Dominance	Post Arousal	Post Valence
Sound	0.19620	0.46847	−0.20359	0.49618	−0.10109
Visible	−0.11445	−0.11022	0.09724	−0.15649	0.05910
IR	−0.13151	−0.17583	0.06240	−0.23523	0.09142
UV	−0.10024	−0.10639	0.09605	−0.15116	0.05522
Temperature	0.26247	−0.06902	0.02411	0.05704	−0.16257
Humid	0.38599	−0.20321	0.09103	−0.01901	−0.25521
Pressure	0.21656	0.15605	−0.02990	0.39601	−0.09800
CO_2_	0.75455	0.25542	−0.00510	0.23169	0.04396
PM_1_	0.39583	0.21699	−0.11200	0.36770	−0.18324
PM_2.5_	0.37961	0.21107	−0.10687	0.35539	−0.19239
PM_10_	0.29759	0.25642	−0.10123	0.34264	−0.17725
Wind	−0.54198	0.13262	−0.20161	−0.05592	−0.10570

## Data Availability

The data presented in this study are available on request from the corresponding author. The data are not publicly available due to institutional protocols.

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
