# Peer review of "Application of DIY Electrodermal Activity Wristband in Detecting Stress and Affective Responses of Students"

_bioengineering, 2024, doi:10.3390/bioengineering11030291_

Round 1

Reviewer 1 Report

Comments and Suggestions for Authors

The research presented in this paper uses electrodermal activity (EDA) sensors to measure and analyze data related physiological and neurological stress underlying cognitive tasks involving students.

Strengths: low cost off-the-shelve sensors were used for measurement.

Weaknesses:  There are frequent repetitions in the paper. The subsection numbering is nonstandard and confusing. A separate Discussion section should be added.

1. A major part of the introduction section and references [1-8] discuss climate change and ecosystems. Similarly, the first two sections in Literature survey and references [13-22] discuss nontechnical topics related to microclimates. Instead, the Literature survey should focus on EDA and EDA-based data analysis.

2. There are frequent repetitions in the paper. For example, the last paragraph of Introduction is repeated verbatim in the Abstract.

3.  Please label (a) and (b) parts in Figure 3.

4. Please use a correct and recognized numbering format in Section 3. The subsections should be numbered 3.1, 3.2, or 3A,3B, etc.

5. Move self-reported measures details (Tables 1 &2) to the Appendix.

6. Correct grammatical mistakes using future tense, e.g., 'Participants will get to rate the difficulty of the task...'

7. Provide more details of Empatica sensors used in the study.

8. Please remove all repetitions from the manuscript. For example, data processing ('EDA data were first resampled to a 4 Hz sampling rate...')  is repeated multiple times in the manuscript (pg. 9, 11, 12).

9. Provide a reference for using Shapley values for judging predictor variables.

10. Combine Figures 7-12 as subfigures in a single figure.

11. There is no Section number for Results and Discussion.

12. Add detailed discussion of the results presented in Tables 4-6. Only 2-3 line discussion is not enough.

13. Summarize the results of Figures 13-23 in one or more tables and discuss accordingly.

14. Add a separate Discussion section to the paper.

15. Follow a recognized numbering format for Results and Discussion.

Comments on the Quality of English Language

English language use is generally fine except for occasional grammatical mistakes.

Reviewer 2 Report

Comments and Suggestions for Authors

Tile: Abbreviations (DIY) should be avoided in the title.

Abstract The way it is currently written presents a summary of the research. Quantitative results and how they support the set goals of the research must be mentioned. Why EEG is mentioned in the abstract (EEG results are not even mentioned in the paper) 

Introduction: Unnecessary long and excessive elaboration of the climate change problem. This research aims at quantifying EDA changes associated with microclimate conditions. The introduction should focus on that and previous attempts to address that aim. Climate change motivation may be briefly mentioned. The introduction must be shortened

Tables 1 and 2: move to the appendix. They make the paper unnecessarily long.

Figure 4 and 5. Please add the units on the vertical axes.

Methods. The DIY was worn on the left wrist. Was handedness taken into account? were all participants right handed?

Table 3: is part of the results section not methods.

Results: in the preliminary statistical  analysis subsection, there is repetition of the methods about EDA data processing. 

Figures 7 to  12. Consolidate into a single multi-panel figure. Having them in the current form makes the paper unnecessarily long

Tables 4 to 6. Report the statistical significance of the correlations. Significant can for instance, be boldened

Discussion EDA changes cannot be specifically attributed to microclimate conditions, physiological phenomena including circadian variations affect EDA. The paper should discuss that in the framework of making sense of the R^2 values.   

An elaborate discussion about the qualitative significance of microclimate condition influence on EDA is needed. Combining results and discussion can be done but I do not see an appropriate interpretation of the results. After reading the introduction and its content on climate change, I was expecting a discussion about the opportunities of measuring EDA to quantitatively characterize the potential adverse consequences of climate change on human physiology

Comments on the Quality of English Language

Finally. There are several typos in the paper. Proof-reading is needed

Reviewer 3 Report

Comments and Suggestions for Authors

The paper analyzes electrodermal activity (EDA) in students to study relationships between environment, health, and stress. There are recommendations as follows.

1) To clearly show the contributions made in this work, it would add impact for the authors to provide an explicit bulleted list highlighting the 3-4 most significant advancements enabled through this research. 

2). To ensure proper formatting, it would be beneficial for the authors to thoroughly verify that the correct manuscript template has been used. Carefully checking elements like page layout, section organization, can help identify any  inconsistencies.

3) Including a few example traces of the electrodermal activity (EDA) signals collected through the wearable sensors would be beneficial for this research.

4) The first occurrence of the abbreviation "DIY" should be expanded to clearly define this term  before continuing to use the shortened DIY form. 

5) To enhance clarity, it is recommended to increase the text size and overall resolution in Figure 6. 

6) While the early data collection provides useful initial findings, expanding the participant pool could further strengthen confidence. The current sample of 5 participants (4 males, 1 female) limits the ability to generalize trends across broader populations. Adding more balanced representation by increasing the total number of participants, and recruiting equal numbers of males and females, would generate higher statistical power and account for a wider range of physiological variability. 

7) To enhance the manuscript's graphical clarity, consider upgrading Figures 13-22 to higher resolution versions.

Round 2

Reviewer 1 Report

Comments and Suggestions for Authors

The previous suggestions have been implemented in the revised version.

Reviewer 2 Report

Comments and Suggestions for Authors

The authors have taking my comments into account in the revised version of the manuscript.

The paper can be accepted in its current form

Reviewer 3 Report

Comments and Suggestions for Authors

The authors have addressed all the previous comments.